# Spinal microcircuits comprising dI3 interneurons are necessary for motor functional recovery following spinal cord transection

Tuan V Bui[1†], Nicolas Stifani[2†], Turgay Akay[2], Robert M Brownstone[2,3,4*]

[1]Department of Biology, Brain and Mind Research Institute, University of Ottawa, Ottawa, Canada; [2]Department of Medical Neuroscience, Dalhousie University, Halifax, Canada; [3]Division of Neurosurgery, Department of Surgery, Dalhousie University, Halifax, Canada; [4]Sobell Department of Motor Neuroscience and Movement Disorders, Institute of Neurology, University College London, London, United Kingdom

**Abstract** The spinal cord has the capacity to coordinate motor activities such as locomotion. Following spinal transection, functional activity can be regained, to a degree, following motor training. To identify microcircuits involved in this recovery, we studied a population of mouse spinal interneurons known to receive direct afferent inputs and project to intermediate and ventral regions of the spinal cord. We demonstrate that while dI3 interneurons are not necessary for normal locomotor activity, locomotor circuits rhythmically inhibit them and dI3 interneurons can activate these circuits. Removing dI3 interneurons from spinal microcircuits by eliminating their synaptic transmission left locomotion more or less unchanged, but abolished functional recovery, indicating that dI3 interneurons are a necessary cellular substrate for motor system plasticity following transection. We suggest that dI3 interneurons compare inputs from locomotor circuits with sensory afferent inputs to compute sensory prediction errors that then modify locomotor circuits to effect motor recovery.

*For correspondence:
R.Brownstone@ucl.ac.uk

[†]These authors contributed equally to this work

**Competing interests:** The authors declare that no competing interests exist.

## Introduction

Like other regions of the central nervous system, the spinal cord is remarkably plastic (**Wolpaw, 2007**; **Grau, 2014**). Such plasticity has been demonstrated, for example, following spinal cord injury, when training can lead to a degree of recovery of spinal locomotor circuits such that stepping movements are restored (**Barbeau et al., 1987**; **Courtine et al., 2009**; **Harkema et al., 2012**; **Hubli and Dietz, 2013**; **Martinez et al., 2013**; **Takeoka et al., 2014**). After complete spinal transection in cats and rodents, a treadmill-training regimen that provides rhythmic sensory input to the spinal cord leads to the re-acquisition of the complex sequence of muscle activation that produces stepping (**Barbeau and Rossignol, 1987**; **Sławińska et al., 2012**). Multiple modalities of sensory input are likely required to promote these sustained changes in spinal circuits, as removing cutaneous inputs degrades the quality of recovery in both cat (**Bouyer and Rossignol, 2003**) and rats (**Sławińska et al., 2012**), and eliminating muscle proprioceptive afferents impairs recovery in mice (**Takeoka et al., 2014**). But in addition to determining the afferent inputs involved, it is necessary to identify the spinal circuits involved in this plasticity in order to understand how the nervous system acquires new motor skills in health and injury.

**eLife digest** Circuits of nerve cells, or neurons, in the spinal cord control movement. After an injury to the spinal cord, the connections between the brain and spinal neurons may be severed, meaning that the spinal circuits can no longer work properly. This loss of communication between the brain and the spinal cord often leads to paralysis below the level of the injury.

There are currently no effective treatments for individuals who have lost the ability to walk following spinal cord injury. However, the spinal cord retains circuits that are sufficient to restore walking and these circuits can be activated with training. That is, rehabilitative training can lead to improvements in movement by promoting spinal cord plasticity – the ability of other neurons in the spinal cord to take over the roles of the severed neurons. By understanding how rehabilitation leads to these improvements following injury, new strategies could be developed to optimize the recovery process.

Previous research showed that spinal neurons called dI3 interneurons are involved in short term adjustments of movement. Could these interneurons also be involved in longer term adaptations?

Bui, Stifani et al. compared normal mice with genetically engineered mice that had dI3 interneurons "removed" from their circuits. This revealed that although dI3 interneurons in mice are integrated with spinal circuits that are involved in walking, they are not necessary for normal walking. Following the severing of the spinal cord, the experimental mice, unlike the normal mice, did not recover the ability to step. Thus, circuits comprising dI3 interneurons are necessary for recovering the ability to move after an injury.

Now that Bui, Stifani et al. have identified this essential circuit, the next step is to investigate how dI3 interneurons promote spinal cord plasticity. Understanding these mechanisms could help to develop therapies that enhance rehabilitation-assisted improvement of movement following spinal cord injury.

One approach towards understanding these circuits would be to study neurons that are interposed between sensory inputs and spinal locomotor circuits. Furthermore, since there is a relationship between short-term adaptation and longer-term plasticity (*Bastian, 2008*), it might be useful to focus on interposed neurons that are known to be involved in adaptive responses. We previously showed that dI3 interneurons (INs), a population defined by expression of the LIM-homeodomain transcription factor *Isl1* (*Liem et al., 1997*), receive multimodal monosynaptic sensory afferent inputs and project to spinal motoneurons. Eliminating glutamatergic output by these neurons led to deficits in motor responses to sensory perturbation: while the mice could place their paws on wire rungs, they were unable to adjust their grasp in response to sensory stimulation provided by increasing the inclination of the rungs. This indicates that a microcircuit involving dI3 INs mediates adaptive changes in motor behaviour (*Bui et al., 2013*). We thus focussed on this population to determine if they have a role in locomotor recovery.

Here, we have considered the position of dI3 INs in spinal microcircuits, and show that they are indeed interposed between sensory inputs and locomotor circuits. We demonstrate that while dI3 INs are not required for normal locomotor function, they are necessary for stable recovery of locomotor activity following spinal cord transection. Specifically, we demonstrate that eliminating glutamatergic output from dI3 INs precludes locomotor recovery after spinal cord transection. Thus, dI3 INs are involved in spinal microcircuits that mediate motor system plasticity.

# Results

## dI3 interneurons are not necessary for locomotor function

We first determined whether dI3 INs are an essential component of spinal locomotor circuits and thus necessary for locomotion. To do so, we genetically eliminated glutamatergic neurotransmission from dI3 INs using dI3$^{OFF}$ (*Isl1$^{Cre/+}$;Slc17a6$^{fl/fl}$*) mice (*Bui et al., 2013*). Within their cages, dI3$^{OFF}$ mice did not reveal obvious locomotor deficits (*Bui et al., 2013*). There was no difference between the weights of control (19.8 ± 2.3 g; n = 14) and dI3$^{OFF}$ mice (19.0 ± 2.7 g; n = 9,

p>0.44). Footprint and inter-limb coordination analysis revealed subtle alterations in locomotion in adults (*Figure 1A*). The hind paws, but not the forepaws, of dl3$^{OFF}$ mice were more widely spaced than those of control mice (*Figure 1B*), and dl3$^{OFF}$ mice had a greater propensity to make coincident contact with the ground with three or four paws (*Figure 1A,C*, and *Video 1*), however inter-limb coordination was similar in dl3$^{OFF}$ and control mice (*Figure 1—figure supplement 1*). During treadmill locomotion (*Video 1*), dl3$^{OFF}$ mice had longer stance times on average, and a steeper relationship between stance duration and cycle period than seen in controls (*Figure 1D,E*). Collectively, the shorter swing phases, increased time with more paws in contact with the treadmill, and more widely spaced hind paws in dl3$^{OFF}$ mice could result from a

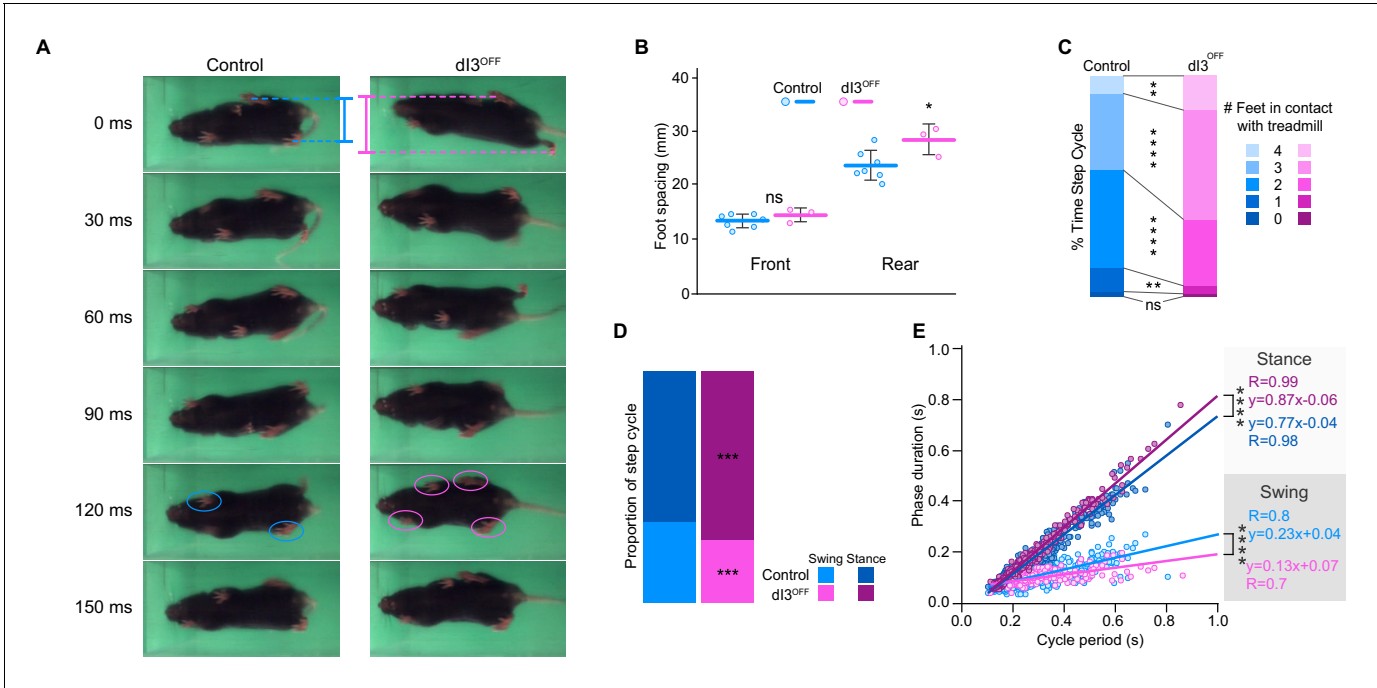

**Figure 1.** dl3 INs are subtly involved in locomotion. (**A**) Footprint snapshots in control (cyan) and dl3$^{OFF}$ (magenta) littermates at 30 ms intervals starting from the onset of the stance of right hindlimb (top) running at 30 cm/s. Coloured lines represent hindlimb feet spacing, circles indicate foot contact. (**B**) Front and rear foot spacing for control (cyan, n = 6) and dl3$^{OFF}$ (magenta, n = 3) animals running between 10 and 50 cm/s. Mean +/− Standard Deviation. (**C**) Proportion of time with indicated number of feet contacting the treadmill belt in same animals running between 10 and 18 cm/s. (**D**) Percentage of swing (light shades) and stance (dark shades) in same animals running between 10 and 18 cm/s. (**E**) Correlation between swing or stance duration and cycle period in same animals running between 9.5 cm/s and 72 cm/s. Each data point represents a single step cycle. Analysis of Covariance (ANCOVA) on slopes. (**B**, **C**, **D**) Two-way ANOVA followed by Sidak post-hoc multiple comparison test. *p<0.05, **p<0.01, ***p<0.001, and ****p<0.0001, ns non-significant.

The following source data and figure supplements are available for figure 1:

**Source data 1.** Related to *Figure 1B*.
**Source data 2.** Related to *Figure 1C*.
**Source data 3.** Related to *Figure 1D*.
**Source data 4.** Related to *Figure 1E*.
**Source data 5.** Related to *Figure 1I*.
**Figure supplement 1.** Interlimb coordination in dl3$^{OFF}$ mice is similar to controls.
**Figure supplement 1—source data 1.** Related to *Figure 1—figure supplement 1A*.

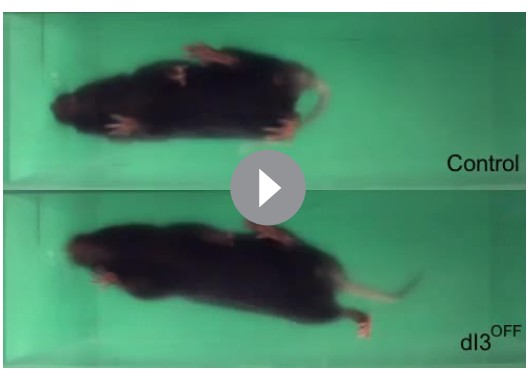

**Video 1.** Foot print recordings in intact animals. Foot print recordings of control and dI3[OFF] animals running at 30 cm/s, recorded at 100 fps and displayed at 15 fps. Animals were recorded separately.

reduction in extensor activity and/or compensation for a reduction in stability. Taken together, these results show that while dI3 INs sculpt hind limb movement, they are not critical for the fundamental rhythm and/or pattern of locomotion.

## dI3 interneurons activate locomotor circuits

We next sought to determine whether dI3 INs, in addition to their projections to motoneurons (*Bui et al., 2013*), also have access to spinal circuits for locomotion. Considering that sensory stimulation can trigger locomotor activity (*Lundberg, 1979*; *Hultborn et al., 1998*) and that dI3 INs can be monosynaptically activated by stimulation of low-threshold sensory afferents (*Bui et al., 2013*), we asked whether dI3 INs could activate spinal locomotor circuits. To do so, we studied sensory-evoked locomotion in control and dI3[OFF] mice, stimulating a sensory (sural) rather than a mixed (tibial) nerve to avoid stimulating motor axons. Furthermore, as sensory afferents may have multiple routes to spinal locomotor circuits, we stimulated the sural nerve, which we previously showed both anatomically and physiologically project directly to dI3 INs (*Bui et al., 2013*), rather than a dorsal root. We thus studied sensory-evoked locomotor activity in neonatal (P1-P3) mice by isolating their spinal cords with the sensory sural nerve in continuity (*Figure 2A*). Given that the Vesicular glutamate transporter 2 (vGluT2) coded by the gene *Slc17a6* is also expressed in high threshold small to medium sized primary afferents responsible for pain, itch, and thermoception (*Brumovsky et al., 2007*; *Lagerström et al., 2010*; *Liu et al., 2010*; *Scherrer et al., 2010*), we focussed on low threshold stimulation, with threshold defined as the lowest amplitude that produced a volley in the proximal dorsal root (A-wave in *Figure 2B*; control: 4.9 ± 6.4 μA, $n$ = 13; dI3[OFF]: 2.9 ± 1.4 μA, $n$ = 12; p>0.29).

In 13 of 14 control spinal cords, stimulation of the sural nerve generated locomotor activity (*Figure 2C,E*), whereas this was only possible in 6 of 12 dI3[OFF] mice (*Figure 2D,E*, p<0.05). Furthermore, the thresholds required for evoking locomotor activity in these 6 dI3[OFF] spinal cords were significantly higher than control thresholds (*Figure 2C–E*), with activity evoked only when stimulation was equal to or greater than the threshold for producing the higher threshold C-wave (in contrast to 5/6 controls that responded to low threshold stimulation, *Figure 2B,F*). In the dI3[OFF] spinal cords in which locomotor activity could be evoked, left-right alternation (control phase: 0.46 ± 0.11; dI3[OFF] phase: 0.53 ± 0.14; p>0.15; *Figure 2—figure supplement 1A*) and locomotor frequency (control: 1.2 ± 0.1 Hz; dI3[OFF]: 1.3 ± 0.2 Hz; p>0.63; *Figure 2—figure supplement 1B*) were similar to control spinal cords. The lack of response to low threshold stimulation in dI3[OFF] spinal cords was not due to sural nerve dysfunction, as low threshold sural nerve stimulation produced a normal volley in the dorsal root (*Figure 2B*), nor was it due to a general inability of afferent stimulation to produce locomotor activity, as dorsal root stimulation was able to evoke locomotion in dI3[OFF] mice (*Figure 2—figure supplement 1C,D*). Thus dI3 IN output is necessary for low threshold-evoked locomotor activity, demonstrating that dI3 INs can activate spinal locomotor circuits (*Figure 2G*).

## dI3 interneurons receive rhythmic inputs from spinal locomotor circuits

To further understand the relationship between dI3 INs and locomotor circuits, we next asked whether there was a reciprocal relationship between them, that is, whether dI3 INs receive inputs from locomotor circuits. During drug-induced locomotor activity in hemisected (*Figure 3A*) or dorsal horn removed (*Figure 3C*) spinal cords from neonatal (P1-P6) Isl1[Cre/+];Thy1-fs-YFP mice ($n$ = 7), whole-cell recordings of upper and lower lumbar dI3 INs revealed cyclic membrane potential depolarisations (*Figure 3B,D* and *Figure 3—figure supplement 1A*), with 27 of 40 (68%) dI3 INs firing

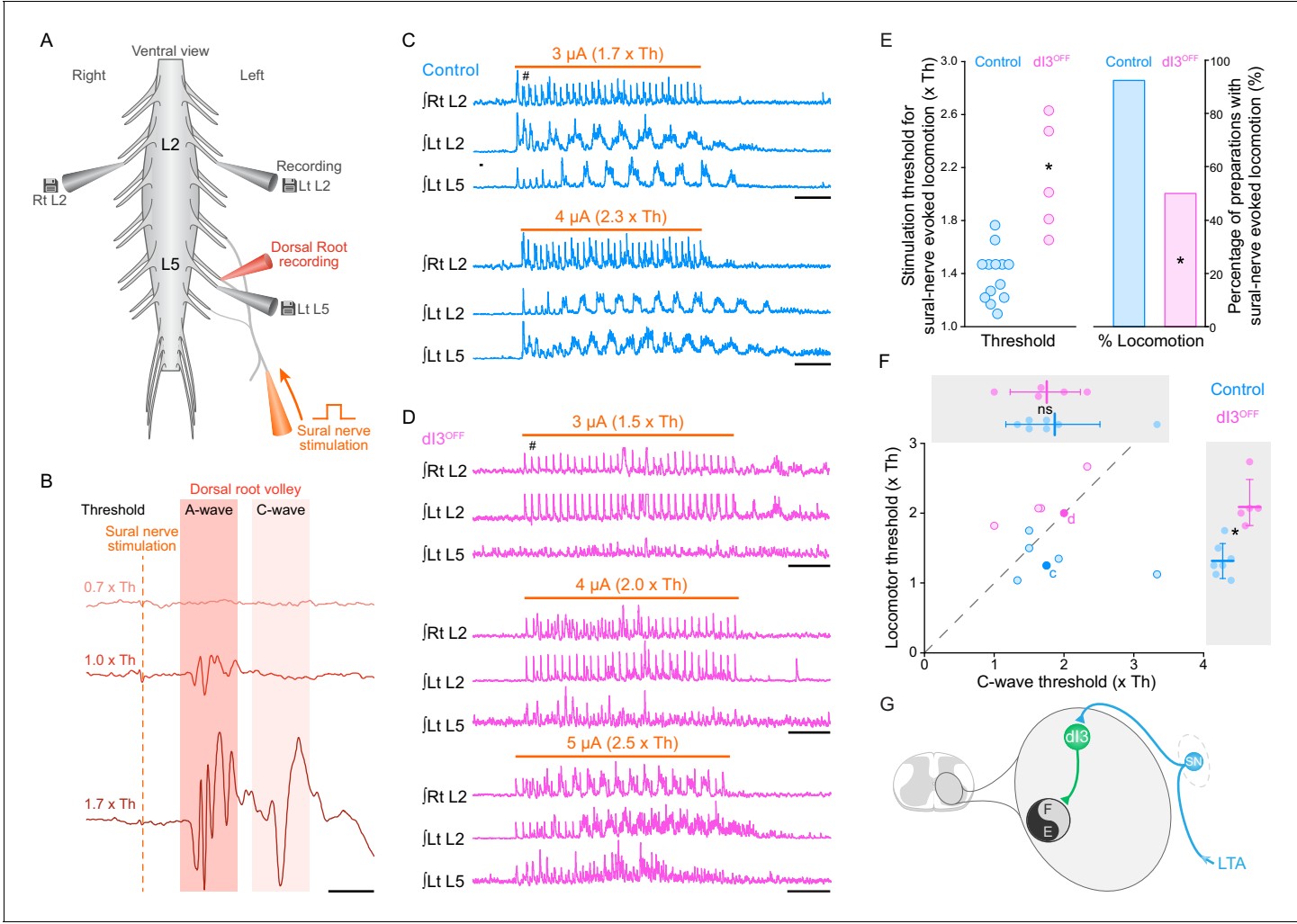

**Figure 2.** dI3 INs activate spinal locomotor circuits. (**A**) Experimental preparation showing isolated spinal cord with sural nerve attached. Sural nerve stimulation (arrow) is used to evoke locomotion, while recording ipsilateral (Left) L5 and bilateral L2 ventral root activity (grey electrodes). Stimulus strength determined by volleys in dorsal root recordings (red electrode). (**B**) Dorsal root potentials in response to sural nerve stimulation (single pulse, 0.25 ms) recorded in ipsilateral L5 dorsal root of a dI3[OFF] spinal cord. Dotted line denotes stimulation artifact. Stimulation strength is in relation to the threshold (Th) to evoke a short-latency A-wave. A-wave threshold was 1.5 µA for this dI3[OFF] spinal cord. (**C**) Rectified and time-integrated ventral root recordings during sural-nerve stimulation (10 s train, 3 Hz, 0.25 ms long pulses, thick orange line) in control mice. Brief voltage deflections in the recordings are stimulation artifacts (#). (**D**) Rectified and time-integrated ventral root recordings during sural-nerve stimulation in dI3[OFF] mice. Note the lack of locomotor activity with stimulation strength ≤2 x Th. Brief voltage deflections in the recordings are stimulation artifacts (#). (**E**) Threshold of stimulation to evoke locomotor activity by sural nerve stimulation (left) and percentage of preparations with sural-nerve evoked locomotor activity (right) in control and dI3[OFF] mice. The threshold of stimulation was not determined for one of the six dI3[OFF] spinal cords in which sural-nerve evoked locomotion was observed. Two-tailed Student's t-test for threshold of stimulation, Fisher's exact test for percentage of successful preparations. (**F**) Relationship between the thresholds for evoking locomotion and the long latency C-wave. Dashed line represents line of unity. Spinal cords in which locomotor but not C-wave thresholds were measured are not shown. There is no difference in C-wave thresholds (inset above), but there is a significant increase in locomotor threshold (right inset) between control (cyan) and dI3[OFF] mice (magenta). dI3 INs corresponding to recordings in panels 2C and 2D are marked by full colored circles and their respective letters. Two-tailed Student's t-test. (**G**) Diagram depicting access of dI3 INs to spinal locomotor circuits. Stimulation of low-threshold afferents (LTA, blue) recruits dI3 INs, which provide drive to spinal locomotor circuits (black and grey circle). *p<0.05, Scale bars, 5 ms (**B**), 2 s (**C, D**).

The following source data and figure supplement are available for figure 2:

**Source data 1.** Related to *Figure 2E*.

**Source data 2.** Related to *Figure 2F*.

*Figure 2 continued on next page*

*Figure 2 continued*

**Figure supplement 1.** When evocable by sural nerve stimulation in dl3[OFF] mice, locomotion is similar to control mice, and when not evocable, locomotion can be initiated by dorsal root stimulation.

rhythmic bursts of action potentials. Raster and polar plots of up to 50 randomly selected action potentials from each of the 27 rhythmically active dl3 INs showed that the majority (22 of 27, 81%) of neurons were active primarily during the extensor phase of the step cycle (cf. 5/27 in flexor phase, *Figure 3—figure supplement 1B*). This extensor-dominant pattern of activity was seen irrespective of whether the interneurons were located in the upper or lower lumbar segments, or medial or lateral grey matter (*Figure 3E* and *Figure 3—figure supplement 1A*). Peak activity of those active during extension was at the midpoint of that phase (*Figure 3F,G*). The dominant extensor phase activity may explain the differences in locomotion between control and dl3[OFF] mice, as elimination of dl3 IN output could result in reduced plantar flexion (physiological extension) (*Bui et al., 2013*) during paw ground contact (*Engberg, 1964*), resulting in reduced weight support and rear track widening (*Donelan et al., 2004*). Together, these data demonstrate that dl3 INs receive inputs – either directly or indirectly – from locomotor circuits, and are predominantly active during the ipsilateral extensor phase of the step cycle.

To determine the nature of this input, we performed voltage-clamp recordings of dl3 INs that were rhythmically active during extensor ($n = 8$) or flexor ($n = 2$) phases. We found that, regardless of their location in the lumbar spinal cord, the post-synaptic currents (PSCs) recorded in 7 out of 8 dl3 INs reversed between −90 and −40 mV, indicating they were inhibitory. In the neurons active in extension, these inhibitory PSCs (IPSCs) were phasic, indicating that they predominantly received rhythmic synaptic inhibition during the flexor phase (*Figure 4A*, and *Figure 4—figure supplement 1A*). Three of these neurons also received some excitatory post-synaptic currents (EPSCs) during the extensor phase. Quantification of the distribution of the IPSCs across the step cycle revealed that the onset and termination of inhibitory input mirrored those of the L2 flexor bursts (*Figure 4B*). When additional brief L2 bursts were present, these bursts also coincided with inhibition of dl3 INs, with the inhibitory input being of longer duration than the abbreviated L2 bursts (*Figure 4C*).

In contrast to the dominant extensor bursting dl3 INs, the two dl3 INs that were active during the flexor phase of the step cycle had no evident rhythmic IPSCs, but were excited during flexion (*Figure 4—figure supplement 1B*), supporting that this subset likely belonged to a different functional sub-population of dl3 INs.

Together, these data indicate dl3 INs are reciprocally connected to spinal locomotor circuits. dl3 INs excite spinal locomotor circuits, and in turn, the majority of dl3 INs are rhythmically inhibited during the flexor phase by these circuits (*Figure 4D*). The timing of this inhibitory input suggests that locomotor circuits are transmitting an inhibitory efference copy to these dl3 INs during the flexor phase of the locomotor cycle.

## dl3 interneurons are necessary for recovery of locomotor function

In light of their interposition between sensory afferents and spinal locomotor circuits, and their involvement in movement adaptation (*Bui et al., 2013*), we next asked whether dl3 INs are involved in microcircuits responsible for plastic changes following spinal cord transection. To do so, we isolated spinal cords from the brain by performing lower thoracic spinal cord transections in control ($n = 22$), and dl3[OFF] ($n = 16$) adult mice. Mice were subjected to regular treadmill training to promote spinal locomotor recovery (*Figure 5A*). We quantified forelimb/hindlimb step ratios as initial estimates of recovered hindlimb locomotor performance over time, counting any (however minimal) forward excursion of the toes as a 'step' (control: $n = 6$; dl3[OFF]: $n = 3$). Step ratios of animals from both groups plateaued by 50 days following transection (*Figure 5C*). But the number of forward toe excursions in dl3[OFF] mice was about half that in control mice (*Figure 5C,D*, and *Video 2*), indicating poor capacity of motor recovery by dl3[OFF] mice following spinal transection.

To assess the quality of locomotor recovery, high-speed kinematic video recordings (*Figure 5B*) were analyzed in the 3 control and 3 dl3[OFF] animals that had anatomically and functionally confirmed complete transections (*Figure 5—figure supplement 1*). The spinal-transected control animals had

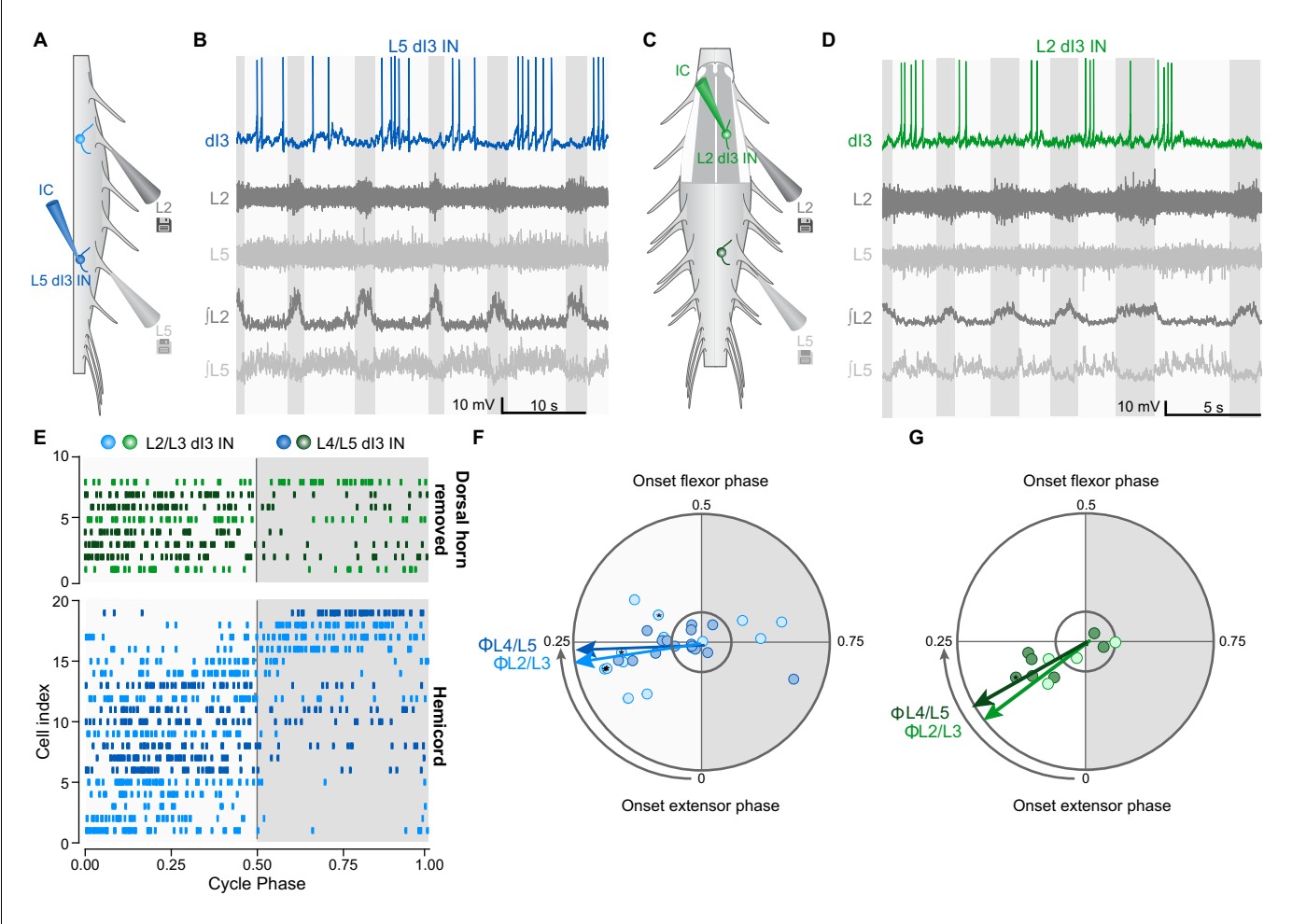

**Figure 3.** dI3 INs are rhythmically active during fictive locomotion. (**A**) Schematic of hemicord preparation indicating the position of recording electrodes as well as L2/L3 (light blue) and L4/L5 (dark blue) dI3 INs. (**B**) Current-clamp recording of L5 dI3 IN rhythmically active during drug-evoked locomotion in hemicord preparation. dI3 IN: whole-cell patch clamp recording. L2, L5: raw ventral root recordings. ∫L2, L5: rectified and integrated ventral root recordings. (**C**) Schematic of L1-L3 dorsal horn removed preparation indicating the position of recording electrodes as well as L2/L3 (light green) and L4/L5 (dark green) dI3 INs. For recordings of L4/L5 dI3 INs, the dorsal horn overlying L4-L6 was removed. (**D**) Current-clamp recording of L3 dI3 IN rhythmically active during drug-evoked locomotion in dorsal horn removed preparation. dI3 IN: whole-cell patch clamp recording. L2, L5: raw ventral root recordings. ∫L2, L5: rectified and integrated ventral root recordings. (**E**) Raster plot of dI3 IN spiking during drug-evoked locomotion. 50 randomly selected spikes (a fewer number of spikes from cells 2, 3, 5, and 14 from hemicord preparations and cell 4 from dorsal horn removed preparation were recorded) are shown for each cell. Locomotor cycles were double-normalized such that a phase of 0 marks the beginning of extensor phase, 0.5 marks the beginning of the flexor phase/end of the extensor phase. Only dI3 INs determined to be rhythmically active are shown (see **F, G**). (**F**) Polar plot summarizing rhythmic activity of dI3 INs in hemicords (n = 13 L2/L3 dI3 INs light blue, and 16 L4/L5 dI3 INs dark blue). Mean phase and angular concentration $r$ calculated from 50 or fewer randomly selected spikes. Each point depicts the average phase at which spikes occurred during the step cycle. The distance away from the centre represents $r$. The inner circle represents $r = 0.24$, which corresponds to significant rhythmicity ($p<0.05$) for 50 randomly selected spikes. Asterisks mark cells where rhythmicity was statistically demonstrated though fewer than 50 spikes were recorded. Arrows depict mean phase (Φ) of dI3 INs whose average phase was during the extensor phase. (**G**) Polar plot summarizing rhythmic activity of dI3 INs in dorsal horn removed spinal cords (n = 4 L2/L3 dI3 INs light green, and 7 L4/L5 dI3 INs dark green). Arrows depict mean phase (Φ) of dI3 INs whose average phase was during the extensor phase.

The following source data and figure supplement are available for figure 3:

**Source data 1.** Related to *Figure 3E*.

**Source data 2.** Related to *Figure 3F*.

**Figure supplement 1.** Examples of rhythmic activity of dI3 INs during fictive locomotion.

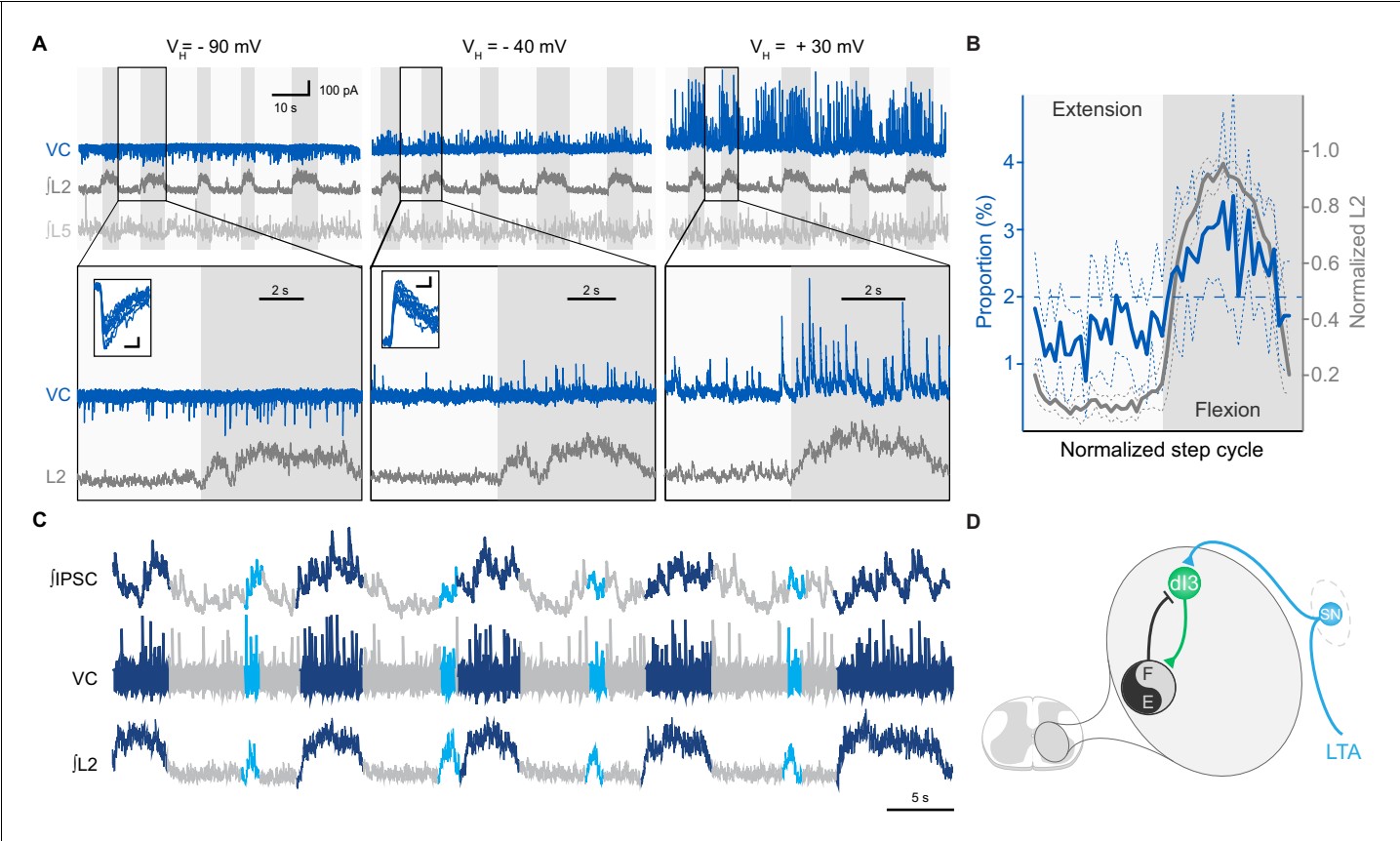

**Figure 4.** dI3 INs receive rhythmic synaptic inputs during drug-evoked locomotion. (**A**) Voltage-clamp recording (VC) of L5 dI3 IN during drug-evoked locomotion at different holding potentials ($V_H$). Bottom two rows are expanded time-scale representation of data in boxes seen in top two rows. Insets depict 10 postsynaptic currents of greatest magnitude within the flexor phase. Scale bars within insets, 5 ms, 25 pA. (**B**) Distribution of IPSCs through the step cycle (n = 6622 IPSCs occurring in 116 step cycles in seven preparations) overlaid with averaged normalised L2 ventral root recordings (means and standard deviations shown). Step cycle was divided into 50 bins. Dashed line at 2% indicates the proportion of IPSCs if they were evenly distributed throughout the step cycle. (**C**) Voltage-clamp recording (VC) at holding potential of −40 mV of L5 dI3 IN during drug-evoked locomotion with brief flexion bursts (cyan) during extension. Bursts of IPSCs were observed during regular (dark blue) and brief flexion bursts as evidenced in integrated IPSCs trace (top trace). (**D**) Diagram showing added inhibition of dI3 INs arising from flexor module of spinal locomotor circuits.

The following source data and figure supplement are available for figure 4:

**Source data 1.** Related to *Figure 4B*.

**Figure supplement 1.** Examples of synaptic inputs to dI3 INs during fictive locomotion.

some recovery of locomotion, although limb movements were reduced compared to their intact counterparts (*Figure 5E–I*, *Figure 5—figure supplement 2*, and *Video 3*). On the other hand, dI3[OFF] mice had minimal horizontal movements of distal hind limb segments (*Figure 5E,G*, and *Figure 5—figure supplement 2C*) and an almost complete absence of vertical excursions (*Figure 5F,I*, *Figure 5—figure supplement 2D*, and *Video 4*). Furthermore, while the kinematic parameters of spinal-transected control animals for the most part were a scaled version of those in intact animals, the parameters in dI3[OFF] animals differed in trajectory (*Figure 5—figure supplement 2E–I*). For example, the knee angle linearly decreased during flexion (when it is normally biphasic) and linearly increased during extension (when it is normally primarily decreasing, *Figure 5I*). That is, while control mice recovered a degree of the complex sequence of muscle activation that produces stepping, the minimal movements of dI3[OFF] mice had linear kinematics with no resemblance to locomotion (*Figure 5H,I*). Furthermore, during unrestrained overground locomotion (data not shown), there was almost a complete absence of hindlimb movement in dI3[OFF] mice, in stark contrast to control mice,

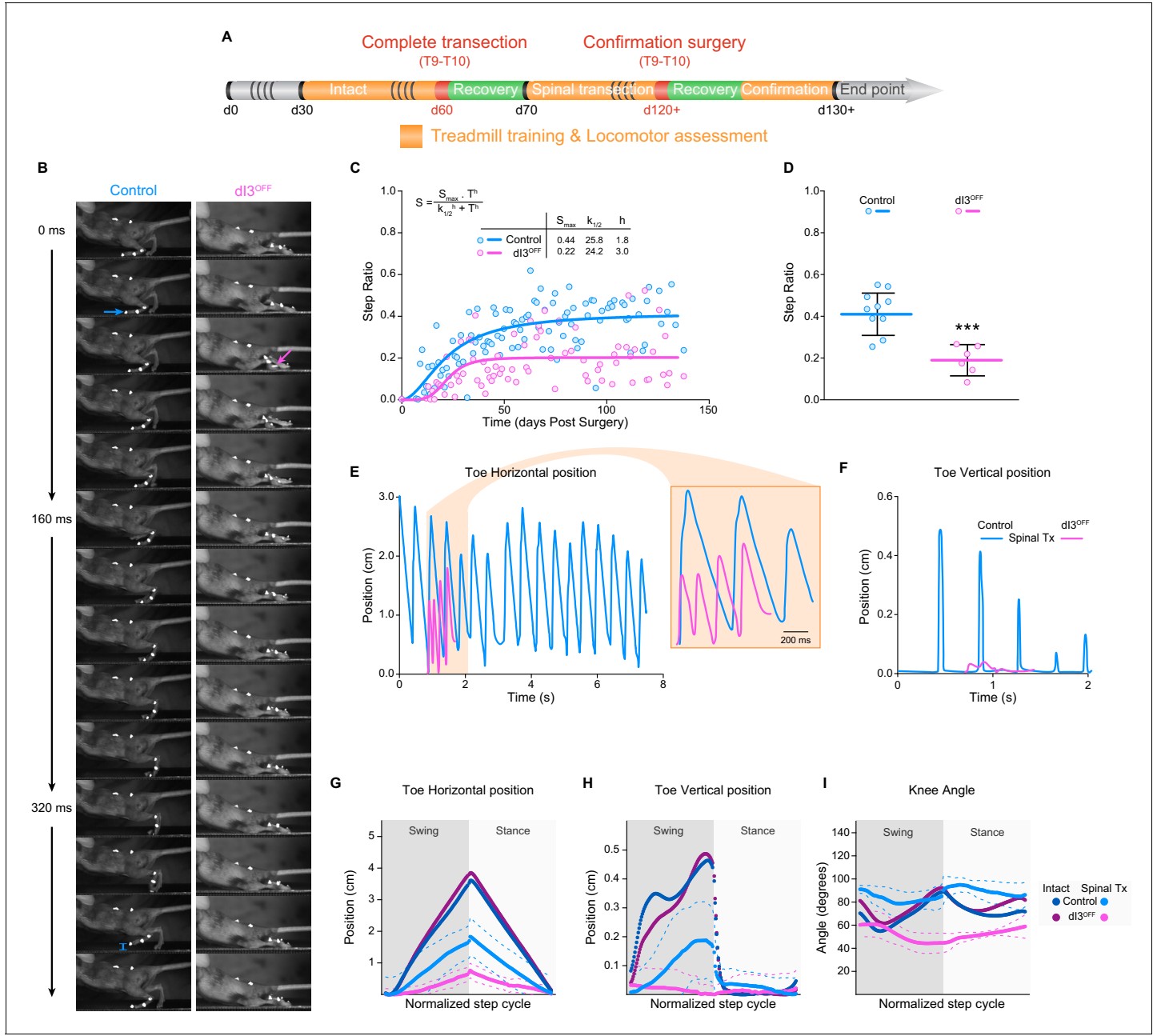

**Figure 5.** dI3 INs are involved in locomotor recovery following complete spinal transection. (A) Experimental paradigm. Orange indicates period of treadmill training and locomotor performance assessments. Green indicates period of recovery following surgical procedures (red). Days (d) from birth are indicated. (B) Kinematic recording snapshots in control and dI3^OFF mice 50 days following complete (and confirmed) lower thoracic spinal transection. Arrows depict proper paw dorsiflexion in control (cyan) and abnormal plantarflexion in dI3^OFF (magenta) mice during stance. Cyan segment at t = 384 ms indicates toe elevation in control, which is absent in dI3^OFF during the swing phase. Pictures at 32 ms intervals. (C) Hindlimb/Forelimb step ratio (S) as a function of time (T) following complete spinal transection in control (n = 6) and dI3^OFF (n = 3) animals with non-linear sigmoidal fit, S = $S_{max} * T^h / (k_{1/2}^h + T^h)$ where $S_{max}$ is the maximum step ratio, h is the Hill slope (recovery rate), and $k_{1/2}$ is the number of days to reach half the maximal step ratio. $R^2$ = 0.53 and 0.23 respectively. (D) Hindlimb/Forelimb step ratio 30 days following complete spinal transection (Tx) in control (n = 10), and dI3^OFF (n = 6) animals. Mean +/− Standard Deviation. Two-tailed Student's t-test (***p<0.0001). (E to I) Toe coordinates along X (horizontal forward-backward axis E,G) and Y (vertical elevation F, H) axes, and knee angle (I) in control and dI3^OFF spinal-transected animals 40 days post-surgery on a treadmill at 7 cm/s. Dashed lines represent standard deviation. Minimum weight-support was provided when necessary to complete the task. In G, H, and I, data from intact animals are shown in dark blue (control) and dark pink (dI3^OFF) for comparison. Multiple t-tests corrected for multiple comparison Holm-Sidak. *p<0.05.

The following source data and figure supplements are available for figure 5:

*Figure 5 continued on next page*

*Figure 5 continued*

**Source data 1.** Related to *Figure 5C*.
**Source data 2.** Related to *Figure 5D*.
**Source data 3.** Related to *Figure 5G*.
**Source data 4.** Related to *Figure 5H*.
**Figure supplement 1.** Confirmation of anatomical and functional spinal isolation.
**Figure supplement 1—source data 1.** Related to *Figure 5—figure supplement 1C*.
**Figure supplement 2.** Locomotor function following isolation of spinal circuits.

suggesting that the minimal movements seen during treadmill walking in dl3[OFF] mice were specific to the treadmill environment.

Taken together, these results illustrate an absence of locomotor recovery in dl3[OFF] animals, and demonstrate the critical role of dl3 INs in driving recovery of locomotor function after spinal transection.

## Discussion

We have shown here that dl3 INs are involved in spinal plasticity, and are integral to recovery of locomotor function following spinal cord transection. They are positioned between multimodal sensory inputs (*Bui et al., 2013*) and spinal locomotor circuits, and have a bi-directional relationship with these locomotor circuits, receiving an efference copy of their activity (*Figure 6A*). We showed previously that elimination of dl3 INs from spinal microcircuits via genetically eliminating their glutamatergic output results in the abolition of short term motor adaptation (*Bui et al., 2013*), and now demonstrate that this mutation also results in the loss of plasticity required for locomotor recovery. Together, we suggest that dl3 INs function to integrate sensory input with motor commands in order to drive motor plasticity at level of the spinal cord (*Figure 6*).

### dl3 IN microcircuits and locomotion

Sensory inputs to the spinal cord are required for the recovery of function following injury (*Dietz et al., 2002*; *Rossignol and Frigon, 2011*; *Takeoka et al., 2014*). Experiments in animal models such as rodents and cats have demonstrated that sensory inputs have access to spinal locomotor circuits, are phasically gated during the step cycle (*Forssberg et al., 1977*), and their stimulation can lead to alterations in the timing and coordination of ongoing locomotor activity (*Duysens and Pearson, 1976*; *Loeb et al., 1987*; *Gossard et al., 1994*; *Stecina et al., 2005*). Furthermore, stimulation of sensory afferents in in vitro isolated spinal cords can be sufficient to activate spinal locomotor circuits (*Bonnot et al., 2002*; *Cherniak et al., 2014*). Therefore, spinal locomotor circuits remain accessible through sensory afferents following the loss of descending inputs.

Activation of sensory afferents by treadmill training is a guiding principle of locomotor

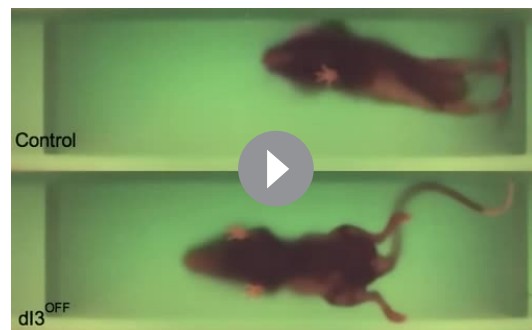

**Video 2.** Foot print recordings following spinal cord transection. Foot print recordings of control and dl3[OFF] animals at 7 cm/s 50 days after spinal transection, recorded at 100 fps and displayed at 15 fps. Animals were recorded separately.

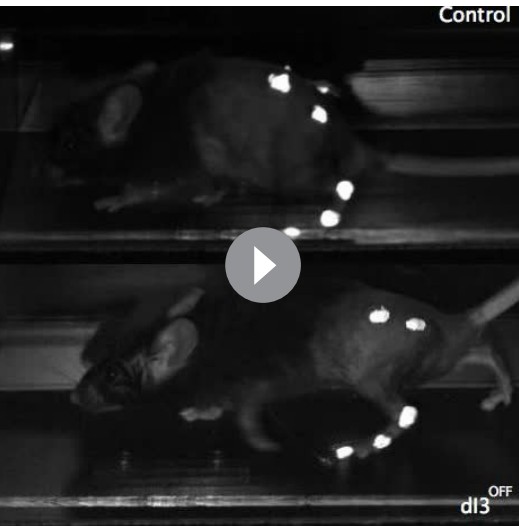

**Video 3.** Kinematic recordings in intact animals. Kinematic recordings of control and dl3OFF animals running at 20 cm/s, recorded at 250 fps and displayed at 15 fps. Animals were recorded separately.

**Video 4.** Kinematic recordings following spinal cord transection. Kinematic recordings of control and dl3OFF animals at 7 cm/s 50 days after spinal transection recorded at 250 fps and displayed at 15 fps. Weight support was provided. Animals were recorded separately.

rehabilitation (*Harkema, 2008*). This strategy has shown that activation of sensory afferents during imposed walking movements retrains spinal locomotor circuits to generate the rhythmic, patterned activation of hindlimb muscles required for locomotion (*Dietz et al., 1995*; *Edgerton and Roy, 2009*; *Rossignol and Frigon, 2011*; *Harkema et al., 2012*). In spinal-transected cats, at least one source of cutaneous afferents from the hindlimb is required to ensure full recovery of treadmill locomotion with plantar foot placement (*Bouyer and Rossignol, 2003*). And following spinal transection in rodents, recovery of hindlimb movements became highly disorganized following pharmacological block of paw cutaneous

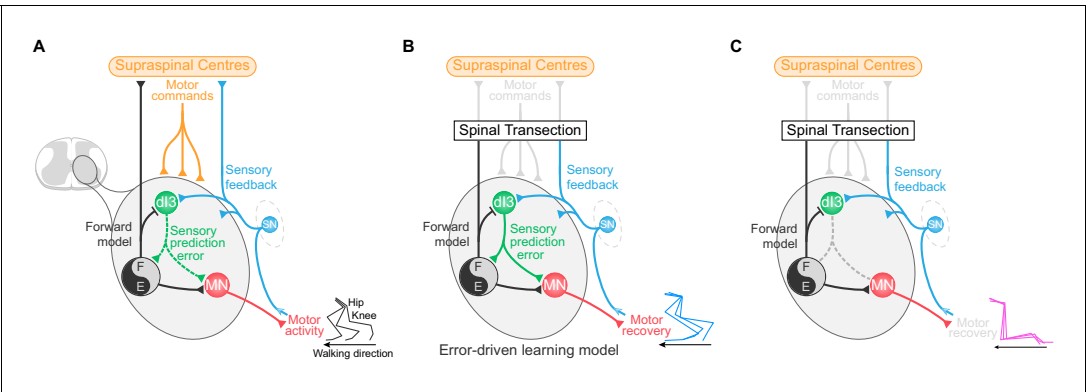

**Figure 6.** dl3 IN-mediated locomotor learning following spinal cord transection. (A) Schematic showing the relationship of dl3 INs (green) to supraspinal centres (orange), sensory input (SN, blue), motoneurons (MN, red), and spinal locomotor circuits (black and grey circle). The types of signals conveyed by each projection are labelled. dl3 IN axons are depicted with dotted lines to demonstrate that they are not necessary for the generation of the normal rhythm and pattern of locomotion. (B) Schematic showing that following loss of connectivity with higher learning centres, locomotor learning is driven by sensory prediction error in a microcircuit centred around dl3 INs as comparators. dl3 INs act as comparators between a forward model from motor circuits and sensory feedback, and the solid projections from dl3 INs indicate that they are required for locomotor recovery following spinal cord transection. (C) Removing the output of dl3 INs from the circuit prevents the comparison of sensory feedback and forward models, and thus precludes functional locomotor recovery.

afferents, indicating the important role of these afferents in functional recovery (*Sławińska et al., 2012*). Proprioceptive afferents also play an important role in recovery following incomplete spinal cord injury (*Takeoka et al., 2014*).

Thus, target neurons responsible for spinal mechanisms involved in sensory-mediated plasticity of locomotor circuits would be expected to receive sensory inputs from a variety of afferent types and project to spinal locomotor circuits. We have shown previously that dI3 INs receive different modalities of sensory inputs (*Bui et al., 2013*), and here we show that they can activate spinal locomotor circuits. Furthermore, we now show that they are necessary for sensory-mediated recovery of function following spinal transection. That dI3 INs receive multimodal sensory inputs could explain why different sensory modalities contribute to locomotor recovery following spinal cord injury and would suggest, perhaps, that the combined inputs from different modalities, through temporal and spatial summation, may be beneficial to training strategies.

The identity of the locomotor neurons excited by dI3 INs remains elusive: in addition to their outputs to motoneurons, dI3 INs also project to as yet unidentified neurons in the intermediate laminae of the cervical and lumbar spinal cord (*Bui et al., 2013*). Unfortunately, although we increasingly understand locomotor circuits (*Kiehn, 2016*), knowledge of the neuronal substrate for activation of these circuits either by descending commands (*Jordan et al., 2008*; *Bretzner and Brownstone, 2013*; *Bouvier et al., 2015*), or by sensory inputs (*Whelan et al., 2000*; *Marchetti et al., 2001*; *Cherniak et al., 2014*) is lacking. Understanding dI3 IN target neurons in the intermediate laminae of the spinal cord may shed light on these questions, and would be a key next step in the determination of the cellular mechanisms of plasticity induced by dI3 INs.

## Possible mechanisms underlying dI3 IN-mediated plasticity

We have shown that dI3 INs are necessary for training-induced recovery of locomotor activity following spinal transection. It is unlikely that this recovery results simply from a loss of afferent input to locomotor circuits, as sensory stimulation – either by increasing stimulation intensity or by number of fibres (by dorsal root stimulation) – can still evoke locomotor activity in dI3$^{OFF}$ mouse spinal cords. We thus suggest that dI3 INs are mediating plastic changes in these circuits. In other words, the re-acquisition of locomotion can be considered to be a low level form of motor learning, in which repeated activity leads to sustained changes in the central nervous system such that spinal circuits below the site of a lesion can produce locomotor activity in the absence of descending motor commands. In this light, it is therefore instructive to consider that circuits that mediate short-term adaptation are those that lead to long-term learning through various mechanisms (*Hirano et al., 2016*). We have previously shown that dI3 INs mediate short-term adaptation in regulating paw grasp in response to changing sensory stimulation (*Bui et al., 2013*). Long-term changes, however, must be accompanied by homeostatic plastic mechanisms that prevent instability induced by positive feedback (*Turrigiano, 1999*; *Desai, 2003*; *Quartarone and Hallett, 2013*). These mechanisms could include, for example, changes in connectivity, synaptic strength, and/or morphology of spinal neurons (*Brownstone et al., 2015*). Such changes have been proposed to underlie spontaneous or training-induced changes in motor output in spinal cord injury patients (*Harkema, 2008*; *Knikou, 2010*; *Dietz, 2012*) and in animal models of spinal cord injury (*Côté and Gossard, 2004*; *Frigon et al., 2009*; *Tillakaratne et al., 2010*; *Martin, 2012*; *van den Brand et al., 2012*; *Houle and Côté, 2013*; *Takeoka et al., 2014*). Similar mechanisms of plasticity may also occur during learning in intact developing and mature spinal cords (*Tahayori and Koceja, 2012*; *Grau, 2014*). Studying such changes at synapses between dI3 INs and their target locomotor circuit neurons may reveal specific mechanisms underlying this plasticity.

## Functional implications

From a circuit perspective, we know from cerebellar studies that motor learning relies on comparator neurons – neurons that compare actual sensory inputs (what did happen, or instructive inputs) with the sensory input predicted by the motor command (what should have happened) (*Bastian, 2006*; *Shadmehr et al., 2010*; *Wolpert et al., 2011*; *Cullen and Brooks, 2015*). Predictive inputs arise from forward models derived from a copy of the motor command – an efference copy. By comparing these two inputs, comparator neurons calculate the 'sensory prediction error,' which is then used to modify circuit function, either for corrective responses or sustained learning (*Shadmehr et al., 2010*;

*Requarth and Sawtell, 2014*; *Brownstone et al., 2015*). Most dI3 INs receive excitatory instructive inputs from a variety of sensory afferents (*Bui et al., 2013*), as well as inhibitory rhythmic input from locomotor circuits. We suggest that this inhibitory input, which mirrors the motor output, is the manifestation of a forward model and is suggestive of a negative image of the expected excitatory sensory input (*Requarth and Sawtell, 2014*). That is, in addition to instructive inputs, dI3 INs receive inputs indicative of a predictive forward model. These two classes of inputs position dI3 INs as comparators between actual and predicted movement, and thus calculators of sensory prediction error. We suggest that this error signal leads to plastic changes in locomotor circuits, mediating long-term learning such as that necessary for locomotor recovery after spinal cord transection (*Figure 6B*).

Within this framework, sensory information provided by training would lead to activation of locomotor circuits, which would then produce a forward model to predict the sensory consequences of the movement. Through training, the sensory prediction error is iteratively calculated by dI3 INs, and would lead to synaptic and/or cellular changes in locomotor circuits, leading to sustained recovery of motor function (*Brownstone et al., 2015*). Therefore, functional removal of dI3 INs from the circuits results in the loss of sensory prediction error signals, and thus prevents the benefit of locomotor training following spinal transection (*Figure 6C*).

Motor learning is distributed across hierarchical control structures, with different levels of the hierarchy functioning together to ensure adaptation and learning (*Kawato et al., 1987*; *Gordon and Ahissar, 2012*). We show that dI3 INs form an intra-spinal closed loop circuit, in which the microcircuits that route sensory information to locomotor circuits are themselves under the influence of the locomotor circuits that they modulate (*Figure 6A*). These closed loops would be nested within other control loops, such as peripheral sensorimotor loops (*Dimitriou and Edin, 2010*) and those situated between spinal motor circuits and descending motor systems (*Arshavsky et al., 1972*; *Hantman and Jessell, 2010*; *Azim et al., 2014*). Together, these nested loops create a hierarchical control system that would optimise motor learning (*Kawato et al., 1987*). That one of these loops may exist in the spinal cord would be important for rehabilitative techniques: targeting intraspinal learning circuits such as those formed by dI3 INs could lead to new strategies to facilitate spinal circuit function in order to improve motor behaviour affected by a number of neurological diseases and injuries.

## Materials and methods

### Animals

All animal procedures were approved by the University Committee on Laboratory Animals of Dalhousie University (protocol 13–143) and conform to the guidelines put forth by the Canadian Council for Animal Care.

Expression of yellow fluorescent protein (YFP) driven by the promoter for the homeodomain transcription factor *Isl1* was obtained by crossing *Isl1*$^{Cre/+}$ (RRID:IMSR_JAX:024242) and *Thy1*-fs-YFP mice to yield *Isl1*-YFP mice. Conditional knockout of *Slc17a6* (vGluT2) in *Isl1* expressing neurons was accomplished by crossing *Isl1*$^{Cre/+}$ mice with a strain of mice bearing a conditional allele of the *Slc17a6* (vGluT2) gene to yield dI3$^{OFF}$ mice as previously described (*Hnasko et al., 2010*; *Bui et al., 2013*). Control animals consisted of Isl1$^{Cre/+}$:*Slc17a6*$^{flox/+}$ and Isl1$^{+/+}$:*Slc17a6*$^{flox/flox}$ mice. No differences were observed between these two control genotypes and animals were thus pooled into a single control group.

### Treadmill locomotion

Limb movement during locomotor behaviour was described by using motion analysis techniques combined with high-speed video recordings of the behaviour (*Akay et al., 2014*). Analysis of footprints during treadmill locomotion (10 cm/s to 50 cm/s) was performed using an Exer Gait XL treadmill (Columbus Instruments) and analyzed with Treadscan Analysis System v4.0 (Clever Sys). Analysis was performed on segments in which the mice maintained their position on the treadmill. For kinematic analysis, mice walked on a treadmill (either custom made by the workshop of Zoological Institute, University of Cologne or an Exer Gait XL from Columbus Instruments) at speeds ranging from 3 to 72 cm/s and recorded with a high-speed camera (Photron PCL R2, Photron; or IL3-100, Fastec Imaging) at a capture rate of 250 frames per second (Fps). The animals were briefly anesthetized

with isoflurane and custom-made three-dimensional reflective markers (~2 mm diameters) were glued onto the shaved skin at the level of the anterior iliac crest, hip, knee (in some cases), ankle, metatarsophalangeal (MTP) joint, and the tip of the fourth digit (toe) of one or both hindlimbs. Joint coordinates were automatically tracked by Vicon Motus software or by custom scripts for ImageJ (*Schneider et al., 2012*) (RRID:SCR_003070) and R (*R Core Team, 2013*) (RRID:SCR_000432). These coordinates were used to compute hip, knee, ankle, and paw angles. For the knee joint, in consideration of movement of the skin over the knee, the actual knee coordinates and angles were calculated geometrically using the length of the femur and tibia. Stance onset was determined by using local maxima of the position of the toe marker in the horizontal plane. Several step cycles were averaged from portion of recordings when animals were producing a steady locomotor output.

## Electrophysiology

Extracellular ventral and dorsal root recordings via suction electrodes were amplified 10,000 X in differential mode, bandpass-filtered (10 Hz to 3 KHz) using a custom-built extracellular amplifier, and acquired at 10 kHz (Digidata 1322A, pClamp nine software, Molecular Devices RRID:SCR_011323). Whole-cell patch-clamp signals were obtained using a MultiClamp 700B amplifier (Molecular Devices) as previously described (*Bui et al., 2013*).

To study sensory-evoked locomotor activity, in-vitro preparations with the sural nerve in continuity with the spinal cord were prepared from *Isl1*-YFP or dI3$^{OFF}$ postnatal (P1-P3) mice. Surgical procedures to isolate the spinal cord were similar to *Bui et al. (2013)* except that the skin of the right hindlimb was dissected and muscles removed to expose and dissect the sciatic and sural nerves. The sural, common peroneal, and tibial nerves were then transected distally. The spinal cords were left to recover for 1–2 hr before recording. Locomotion was induced by dorsal root or sural nerve stimulation using a 10 s long train of 250 µs pulses at 3 Hz using a Grass Technologies S88 square pulse stimulator (Natus Neurology Inc.). The presence of more than four successive rhythmic bursts within the stimulation train in at least one ventral root was used to indicate the presence of locomotor-like activity.

To record rhythmic inputs to dI3 INs during fictive locomotion, hemisected spinal cords were prepared from *Isl1*-YFP postnatal (P1-P6) mice as previously described (*Bui et al., 2013*). Following anaesthesia with xylazine and ketamine, mice were decapitated. Their spinal cords were isolated by vertebrectomy in room temperature recording ACSF (in mM: NaCl, 127; KCl, 3; NaH$_2$PO$_4$, 1.2; MgCl$_2$, 1; CaCl$_2$, 2; NaHCO$_3$, 26; D-glucose, 10). Ventral and dorsal roots were dissected as distally as possible, and spinal cords were pinned with the ventral side up. A surgical blade was used to make a longitudinal incision down the midline (in a rostro-caudal direction). The hemicords were then allowed to equilibrate in room temperature recording ACSF for at least one hour, then pinned, medial-side up to a base of clear Sylgard (Dow Corning) in a recording chamber and perfused with circulating room temperature recording ACSF. Ventral roots were placed in suction electrodes (A-M Systems Inc.). In a subset of experiments examining the activity of dI3 INs during drug-evoked locomotion, bilateral dorsal cords were removed from segments L1 to L3 or from segments L4 to L6. Spinal cords were pinned on their sides and insect pins were used to trace a line followed by a surgical blade to section the spinal cord.

Fictive locomotion was elicited by application of NMDA (5 µM), serotonin (10 µM) and dopamine (50 µM) (*Jiang et al., 1999*). Data from spinal cords in which synchronous activity was produced in flexor- and extensor-dominant ventral roots were excluded. Circular statistics (*Zar, 1996*) using 50 randomly selected spikes for each dI3 IN were used to determine the phasic relationships between dI3 IN spiking and ventral root bursting during drug-induced locomotion.

Whole-cell patch-clamp electrodes were filled with an internal solution containing (in mM: K-methane-sulphonate, 140; NaCl, 10; CaCl$_2$, 1; HEPES, 10; EGTA, 1; ATP-Mg, 3; GTP0.4; pH 7.2–7.3 adjusted with KOH; osmolarity adjusted to ~295 mOsm with sucrose). In the hemisected cords, care was taken to record from cells that were in the middle of the ventral-dorsal axis of the hemicord to avoid motoneurons. In order to avoid recording from neurons close to the central canal (possibly sympathetic preganglionic neurons), neurons that were near the surface were also avoided.

Most dI3 INs exhibited rhythmic, alternating phases of membrane depolarization, which in some cases were accompanied by firing of action potentials, and quiescent hyperpolarization. In the cells where cyclical change of membrane potential was not accompanied by firing of action potentials,

application of a bias depolarizing current led to phasic firing of action potentials during periods of membrane depolarization.

The locomotor cycle was double-normalized (Kwan et al., 2009) such that each cycle was divided into extensor and flexor phases, with 0 to 0.5 spanning the onset to the termination of the extensor phase and 0.5 to 1 spanning the flexor phase of each cycle. Polar plots of rhythmic activity were used for analysis of the rhythmic activity of each cell. The angle represents the mean phase ($\Phi$) whereas the distance from origin is a measure of concentration around the mean phase, which we refer to here as $r$. After calculating the cosine and sine of the phase of each spike, $\Phi$ was calculated as the average cosine and sine of the spikes and the angular concentration, $r$, was calculated as the square root of the sum of squares of the average cosine and average sine (Zar, 1996). Thus if the neuron fires once per cycle at precisely the same phase, then $r = 1$. Conversely, if there were a random distribution of firing across the step cycle, then the average sine and cosine would both be 0, and $r = 0$ (i.e. all spikes are uniformly dispersed across the step cycle). The inner circle represents an $r$ value of 0.24, as for 50 randomly-selected spikes, values above this threshold indicate that cells were rhythmically active ($p < 0.05$).

To determine the rhythmicity of inputs to dI3 INs, unitary PSCs were extracted from voltage-clamp recordings at $-40$ mV. Outward IPSCs and inward EPSCs were detected in Clampfit using a template. The phase of each IPSC and EPSC was calculated and circular statistics were applied as above to determine whether dI3 INs received rhythmic inputs. The proportion of IPSCs that occurred within a certain phase of the step cycle was determined by calculating the proportion of IPSCs within each of 50 equal bins across the step cycle.

## Spinal cord transection

Complete transections were performed at T9-T10 under isoflurane anaesthesia. Animals were individually housed, given analgesics for 3 days, and allowed to recover for at least one week prior to further testing. Animals were monitored twice daily and bladders expressed manually. Humane endpoints were defined as self-mutilation, improper feeding, decreased grooming, ataxia, or a loss of body weight >20%. Completeness of spinal transections was confirmed by a second surgery as described below.

Locomotor training was provided at least once per week (average twice per week). Animals were allowed to habituate to the treadmill environment before recordings ranging from 20 to 40 s at speeds between 3 and 7 cm/s. When required to ensure the completion of the task, animal support was provided by holding the animal in a horizontal position while maintaining permanent contact with the treadmill belt. 5 to 10 recordings were performed per sessions.

Animals showing regular locomotor performance were selected for spinal transection. In sterile conditions, animals were anesthetized with isoflurane to the point of loss of hindlimb reflexes, then shaved and disinfected using Chlorexidine, 70% isopropyl alcohol and 10% betadine from the neck to the lumbar region, and placed on a heating pad. A Tegaderm film (Ref 70200749300, 3M) was placed for stability and to ensure sterile conditions throughout the procedure. An incision was made, paraspinal muscles were incised and spread to expose the T9-T10 laminae, and laminectomy was performed to expose the spinal cord. Double incisions were performed on the rostral and caudal portions of the exposed cord. A Pasteur pipet melted into a custom hook was used to insert underneath the cord and remove the sectioned segment. A piece of sterile Absorbable Hemostat Surgifoam (Ref 63713-0019-75, Ethicon) was placed into the created spinal space. Deep and superficial muscles were sutured with Chromic Gut 6–0 sutures (Ref 796G; Ethicon), and skin with 6–0 Ethilon nylon monofilament suture (Ref 1856G; Ethicon). Tissue was kept moist at all times using sterile saline solution.

Animals, individually housed until suture removal about 10 days later, were allowed to recover for at least one week prior to further testing. They were monitored twice per day and bladders were expressed manually as necessary. Animal weight was monitored daily. Analgesics were administered sub-cutaneously for 3 days after the surgery (buprenorphine: 0.05 mg/kg working solution, 3 µg/ml twice daily; and ketoprofen: 5 mg/kg working solution, 1 mg/ml once a day). Antibiotic (enrofloxacin: 5 mg/kg, working solution 0.25 mg/ml) was administered sub-cutaneously at the time of surgery then provided in drinking water (400 mg/l) until complete healing. Animals were provided with accessible food, water, and soft bedding.

We confirmed complete transection (*Figure 5—figure supplement 1A,B*) by proceeding with a second transection at least 40 days after the first in 11 control and 6 dI3[OFF] mice (7 and 5 survived, respectively). Animals were excluded (4 control and 2 dI3[OFF] mice) from the analysis if the second surgery induced a locomotor deficit (*Figure 5—figure supplement 1C*), indicating incomplete initial transection.

### Step ratio analysis

In order to assess locomotor function recovery, we quantified the ratio of hindlimb to forelimb steps. A hindlimb step was counted when there was any forward movement of the toes. Ventral treadmill video recordings were analyzed using the ImageJ Cell counter plugin. An average of 40 forelimb steps were counted per recording. The number of rear steps was averaged between both hindlimbs. The ratio was calculated as the average number of hindlimb steps divided by the number of forelimb steps.

### Kinematic recording analysis

For kinematic analysis, were analyzed using a custom script in R (https://github.com/nstifani). The following packages were used: tcltk, zoo, and rgl, all available from the Comprehensive R Archive Network (CRAN repository http://cran.r-project.org). Data recorded from intact control ($n = 12$) and dI3[OFF] ($n = 6$) animals running at 10 to 60 cm/s. Left and right hind limbs were recorded separately and processed in parallel. A total of 382 control and 172 dI3[OFF] step cycles were analyzed. Subsequently, step cycles (swing and stance) were normalized to durations of 100 data points (equivalent to a normalised step cycle duration of 400 ms). Measured variables were averaged per animal and across recordings. Following transection and recovery, data recorded from spinalized controls ($n = 3$) and dI3[OFF] ($n = 3$) animals at speeds between 3 and 7 cm/s were added to the recorded data from intact animals. Steps cycles (control: $n = 46$; dI3[OFF]: $n = 16$) were normalized as described above.

## Acknowledgements

We thank Larry Jordan, Tom Jessell, and Marco Beato for helpful comments, Angelita Alcos and Nadia Farbstein for technical assistance, Tom Hnasko for the *Slc17a6* (vGluT2) conditional knockout mice, Frederic Bretzner, Pratip Mitra, Philippe Magown, and Izabela Panek for discussion and comments, Meggie Reardon for assistance in developing the sural nerve attached spinal cord preparation, Amrit Sampalli and Islay Wright for technical support and mathematical input. TVB was supported by a Nova Scotia Health Research Foundation post-doctoral fellowship, a CIHR Research Fellowship and a NSERC Discovery Grant (RGPIN-2015–06403). This research was undertaken, in part, thanks to funding to RMB from the Canada Research Chairs program, and was funded by the Canadian Institutes of Health Research (FRN 79413).

## Additional information

### Funding

| Funder | Grant reference number | Author |
| --- | --- | --- |
| Canadian Institutes of Health Research | Operating grant, FRN 79413 | Robert M Brownstone |
| Nova Scotia Health Research Foundation | Post-doctoral fellowship | Tuan V Bui |
| Natural Sciences and Engineering Research Council of Canada | Discovery grant, RGPIN-2015-06403 | Tuan V Bui |
| Canada Research Chairs | Research Chair | Robert M Brownstone |
| Canadian Institutes of Health Research | Fellowship | Tuan V Bui |

The funders had no role in study design, data collection and interpretation, or the decision to submit the work for publication.

### Author contributions
TVB, NS, TA, Conception and design, Acquisition of data, Analysis and interpretation of data, Drafting or revising the article; RMB, Conception and design, Analysis and interpretation of data, Drafting or revising the article

### Author ORCIDs
Tuan V Bui, http://orcid.org/0000-0003-0024-1544
Nicolas Stifani, http://orcid.org/0000-0001-8584-9561
Robert M Brownstone, http://orcid.org/0000-0001-5135-2725

### Ethics
Animal experimentation: All animal procedures were approved by the University Committee on Laboratory Animals of Dalhousie University (protocol 13-143) and conform to the guidelines put forth by the Canadian Council for Animal Care.

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
