## [Decision Letter]

Thank you for submitting your article "Spinal microcircuits comprising dI3 interneurons are necessary for motor recovery following spinal cord transection" for consideration by *eLife*. Your article has been favorably evaluated by Eve Marder (Senior Editor) and three reviewers, one of whom, Ole Kiehn (Reviewer #1), is a member of our Board of Reviewing Editors, and another is Serge Rossignol (Reviewer #2).

The reviewers have discussed the reviews with one another and there is general enthusiasm about the work; however, there are issues that need to be addressed in a revision. The Reviewing Editor has drafted this decision to help you prepare a revised submission. Included are also the individual reviews.

Summary:

The manuscript by Bui et al. reports the functional role of a selected group of interneurons (dI3) in mouse normal locomotion and recovery after spinal cord transection. This work is based on elegant mouse genetic experiments to visualize these neurons through their specific transcription factor expression and on the results of genetic ablation of dI3 cells. Using kinematic analysis of locomotion, motor behavioral tests and electrophysiological measurements, Bui et al. show that dI3 cells receive a direct sensory input from hindlimbs and play a moderate role in normal locomotion. The function of dI3 cells becomes much more prominent after complete transection of the spinal cord: thus, despite regular treadmill training, dI3OFF mice show gross impairment in regaining motor behavior. The authors explain their findings by assigning a key role of the dI3 INs in a learning spinal circuit suggesting that dI3 neurons are important during the critical phase of network plasticity that ensures the quality and extent of recovery after injury. This work is of general interest. The results have translational value to the human condition and deal with the thorny issue of recovery after injury and the relative contribution of neurorehabilitation. While the authors' work is of high quality, innovative and well illustrated there are points that should be resolved to improve the presentation and discuss alternative explanations for the rehabilitation.

Essential revisions:

1) It is not so clear if the authors believe that it is only mechanoreceptors in the skin that are acting on the dI3 neurons or if other sensory modalities e.g. from muscle spindles or tendon organs also can reach the dI3 neurons. Muscle proprioception may be involved in the reconfiguration that happens after spinal cord injury. Some justification for the entire focus on the sural nerve stimulation in the study is needed. This issue becomes particularly important when the authors start to talk about the possibility to evoke locomotion with dorsal root stimulation in the absence of transmission in the dI3 neurons. The authors should somehow clarify these issues when they introduce the sural nerve stimulation.

Also locomotion is induced in the control at 1.7th which clearly evokes a C wave as shown in Figure 2. The conclusion really hinges on the fact that in dI3^OFF^, stimulation below 2.5Th does not generally evoke locomotion. Figure 2 shows that the distinction is somewhat marginal. It is unfortunate that there is no data shown for stimulation of locomotion at 1Th since already C waves are evoked at 1.7Th, the lowest level of stimulation evoking locomotion in the control. Please discuss this also in relationship to stimulation of other afferents.

2) Subsection “dI3 interneurons are not necessary for locomotor function”, first paragraph; Figure 1 shows that indeed the interlimb distance is different between control and dI3^OFF^ mice. However, it is quite apparent that the two mice have different sizes and that the interlimb distance may be more related to the facts that the mice are simply bigger. Has this been considered? Is the average weight of the control and dI3^OFF^ mice similar? Please provide data that consider this issue.

3) The whole-cell recordings during fictive locomotion are of particular interest in showing alternation of excitation and inhibition in dI3 neurons (Figure 4). It is clear from the panel in A that there is an alternating rhythmic excitation and inhibition, which is seen when the membrane potential is held at +30 mV. There is no mention of this in the text; instead it is said consistently that these neurons only receive a rhythmic inhibition. The authors need to clarify or at least mention this in the text and rectify the description; maybe also describe this in relation to the so-called "balanced network" theory (Berg & Hounsgaard).

4) It is not clear what the authors imply by the word learning. It was shown long time ago that after an acute spinalization, cats can express spinal locomotion without having to relearn the locomotor pattern. So, the circuits are present; evident from the pharmacological stimulation. If no pharmacological stimulant is given, the recovery process of spinal locomotion depends on re-establishing a proper synaptic excitation/inhibition through sensory stimulation as well as intrinsic neuronal changes to progressively reactivate the circuitry. So is the locomotor deficit in spinal dI3^OFF^ mice a deficit in learning capability or a shutting down of important sensory feedback through dI3 INs on the CPG?. In the end, what the authors provide is the evidence that inputs through a class of IN is important for the expression of spinal locomotion. Learning through training implies that the approximate movements can be somewhat spontaneously generated. If there is no movement, training will be of little use. This issue should be discussed.

5) The Discussion is too long and contains a large amount of speculative arguments which may not be totally relevant to the reader. Thus, to interpret their results the authors claim to adopt "cerebellar models of motor learning" (subsection “Possible mechanisms underlying dI3 IN -mediated plasticity”, first paragraph) that are then expanded to "comparator neurons calculate the sensory prediction error" and to electric organ discharge of mormyrid fish (second paragraph of the aforementioned subsection). The Discussion should be cut down substantially and avoid excessive speculations that are not close to the actual data. When implementing these changes the authors should address the alternative explanations to the learning (see above).

6) It is not clear if dI3^OFF^ spinal cords show any change in structure and cell composition? While the authors have done a lot of electrophysiological measurements, it is not clear if there are any discernible alterations in neuronal/glial composition of the genetic model. This information is not apparent even in the earlier publication (Neuron 2013) by the same group. Of course, while reflexes and network activities may rely on the actual operation of a limited number of circuit components, after transection even an apparently modest histological change may have profound consequences on function. Some discussion of this issue is needed.

---

## [Author Response]

*Essential revisions:*

*1) It is not so clear if the authors believe that it is only mechanoreceptors in the skin that are acting on the dI3 neurons or if other sensory modalities e.g. from muscle spindles or tendon organs also can reach the dI3 neurons. Muscle proprioception may be involved in the reconfiguration that happens after spinal cord injury. Some justification for the entire focus on the sural nerve stimulation in the study is needed. This issue becomes particularly important when the authors start to talk about the possibility to evoke locomotion with dorsal root stimulation in the absence of transmission in the dI3 neurons. The authors should somehow clarify these issues when they introduce the sural nerve stimulation.*

This study is not intended to dissociate the types of afferents that are responsible for recovery of function, but rather the spinal circuits that mediate this recovery. Interestingly – and possibly importantly – dI3 INs receive multimodal sensory inputs (Bui et al., 2013). That is, they integrate information that arises from different afferent types. One type includes low threshold cutaneous mechanoreceptors. We have now stressed this point (subsection “dI3 IN microcircuits and locomotion”, third paragraph), and hope that this is now clear in the manuscript.

There is no doubt that primary afferents have many synaptic targets in the cord. In our 2013 paper, we showed reduction of disynaptic reflex responses in adult mice when stimulating the sural (primarily cutaneous) nerve. Stimulation of the tibial nerve, a mixed nerve, led to less clear-cut results because of obscuration by monosynaptic reflexes. That is, sural nerve stimulation resulted in clearer responses, likely because of the reduced number of pathways between these afferents and the motoneurons.

In parallel to this, it is plausible that we would have had similar results in this study regardless of which afferents we stimulated. The reason for choosing the sural nerve is one of signal to noise, because the quantity of afferents stimulated would be lower – and yet effective in eliciting activity in controls – and so we would be more likely to get a “cleaner” result (which we did). So we don’t think there is anything necessarily magical about either the sural nerve or cutaneous afferents in this sense, but this paradigm was useful to demonstrate that dI3 INs can activate locomotor circuits. We have clarified the text surrounding our choice to use the sural nerve (subsection “dI3 interneurons activate locomotor circuits”, first paragraph).

*Also locomotion is induced in the control at 1.7th which clearly evokes a C wave as shown in Figure 2. The conclusion really hinges on the fact that in dI3^OFF^, stimulation below 2.5Th does not generally evoke locomotion. Figure 2 shows that the distinction is somewhat marginal. It is unfortunate that there is no data shown for stimulation of locomotion at 1Th since already C waves are evoked at 1.7Th, the lowest level of stimulation evoking locomotion in the control. Please discuss this also in relationship to stimulation of other afferents.*

We apologise for the confusion induced by the assumption that Figure 2 are from the same preparation. We don’t think the difference is marginal as illustrated in 2F andsupported by statistical analysis. The C-wave threshold differs between animals, and hence in that figure we plotted locomotor threshold vs. c-wave threshold for each animal. None of the mutants had rhythmic activity when stimulation was below the line of unity, whereas only one control was above the line of unity. We have now marked the points in Figure 2 that correspond to the preparations shown in Figure 2, and trust that this clarifies the issue.

*2) Subsection “dI3 interneurons are not necessary for locomotor function”, first paragraph; Figure 1 shows that indeed the interlimb distance is different between control and dI3^OFF^ mice. However, it is quite apparent that the two mice have different sizes and that the interlimb distance may be more related to the facts that the mice are simply bigger. Has this been considered? Is the average weight of the control and dI3^OFF^ mice similar? Please provide data that consider this issue.*

We agree that this is an important point. While we previously showed that adult dI3^OFF^ mice are similar in weights to controls, we have now included this in this manuscript as well (subsection “dI3 interneurons are not necessary for locomotor function”).

*3) The whole-cell recordings during fictive locomotion are of particular interest in showing alternation of excitation and inhibition in dI3 neurons (Figure 4). It is clear from the panel in A that there is an alternating rhythmic excitation and inhibition, which is seen when the membrane potential is held at +30 mV. There is no mention of this in the text; instead it is said consistently that these neurons only receive a rhythmic inhibition. The authors need to clarify or at least mention this in the text and rectify the description; maybe also describe this in relation to the so-called "balanced network" theory (Berg & Hounsgaard).*

In Figure 4, the reversal potential of the PSCs, whether during extension or flexion, is between -90 and -40 mV. This indicates that these are inhibitory PSCs. Excitatory PSCs would be expected to be inward at -40 mV, and if there are any of these, they are essentially indistinguishable from noise. We do mention in the text that a few cells received excitation (subsection “dI3 interneurons receive rhythmic inputs from spinal locomotor circuits”, second paragraph) but have no clear-cut evidence of a Berg and Hounsgaard “balanced” network. We trust that the revised text is clearer.

*4) It is not clear what the authors imply by the word learning. It was shown long time ago that after an acute spinalization, cats can express spinal locomotion without having to relearn the locomotor pattern. So, the circuits are present; evident from the pharmacological stimulation. If no pharmacological stimulant is given, the recovery process of spinal locomotion depends on re-establishing a proper synaptic excitation/inhibition through sensory stimulation as well as intrinsic neuronal changes to progressively reactivate the circuitry. So is the locomotor deficit in spinal dI3^OFF^ mice a deficit in learning capability or a shutting down of important sensory feedback through dI3 INs on the CPG? In the end, what the authors provide is the evidence that inputs through a class of IN is important for the expression of spinal locomotion. Learning through training implies that the approximate movements can be somewhat spontaneously generated. If there is no movement, training will be of little use. This issue should be discussed.*

Thank you for this comment. We have revised the text to raise this important point (subsection “Possible mechanisms underlying dI3 IN-mediated plasticity”).

Regarding the issue of the use of the word “learning,” which we use late in the Discussion, we think this is partly an issue of semantics. Rossignol, for example, has provided evidence of locomotor circuit plasticity (“learning”) following hemisection. Learning involves the acquisition of a skill (for example) by training, and we would argue that this is what is happening following spinalisation, whether in the cat or the mouse or any other animal. We don’t know the mechanism by which training leads to “re-emergence” (?) of the activity of locomotor circuits. (Is it LTP? synaptic scaling? Etc.), but raise possibilities.

We trust that our changes are satisfactory.

*5) The Discussion is too long and contains a large amount of speculative arguments which may not be totally relevant to the reader. Thus, to interpret their results the authors claim to adopt "cerebellar models of motor learning" (subsection “Possible mechanisms underlying dI3 IN -mediated plasticity”, first paragraph) that are then expanded to "comparator neurons calculate the sensory prediction error" and to electric organ discharge of mormyrid fish (second paragraph of the aforementioned subsection). The Discussion should be cut down substantially and avoid excessive speculations that are not close to the actual data. When implementing these changes the authors should address the alternative explanations to the learning (see above).*

We have shortened the Discussion. We would argue that the crux of this paper is the evidence that dI3 INs are positioned as comparator neurons based upon theoretical frameworks established for other areas of motor learning in the nervous system, most notably the cerebellum. Therefore, while we have substantially reduced the amount of text devoted to this discussion, we think it still of value to bring up this concept.

*6) It is not clear if dI3^OFF^ spinal cords show any change in structure and cell composition? While the authors have done a lot of electrophysiological measurements, it is not clear if there are any discernible alterations in neuronal/glial composition of the genetic model. This information is not apparent even in the earlier publication (Neuron 2013) by the same group. Of course, while reflexes and network activities may rely on the actual operation of a limited number of circuit components, after transection even an apparently modest histological change may have profound consequences on function. Some discussion of this issue is needed.*

We have not experimentally addressed the morphological and anatomical changes in this study, but focused on identifying circuits involved in recovery of function. We discuss possible changes in connectivity, synaptic strength, and morphology in spinal neurons that could lead to recovery (subsection “Possible mechanisms underlying dI3 IN-mediated plasticity”). However, given that the reviewers would like us to shorten our Discussion, we have decided not to delve into neuronal-glial compositions.